# Teacher and Student Views on the Feasibility of Peer to Peer Education as a Model to Educate 16–18 Year Olds on Prudent Antibiotic Use—A Qualitative Study

**DOI:** 10.3390/antibiotics9040194

**Published:** 2020-04-19

**Authors:** Cliodna A. M. McNulty, Carla L. Brown, Rowshonara B. Syeda, C. Verity Bennett, Behnaz Schofield, David G. Allison, Nick Francis

**Affiliations:** 1Primary Care and Interventions Unit, Public Health England, Gloucester GL1 1DQ, UK; carlalbrown@gamedrlimited.com (C.L.B.); rowshonara.syeda@phe.gov.uk (R.B.S.); 2Division of Population Medicine, Cardiff University, Cardiff CF14 4XN, UK; BennettCV@cardiff.ac.uk; 3Faculty of Health and Applied Sciences, University of West of England, Bristol BS16 1QY, UK; behnaz.schofield@uwe.ac.uk; 4Division of Pharmacy & Optometry, University of Manchester, Manchester M13 9PT, UK; David.Allison@manchester.ac.uk; 5School of Primary Care, Population Sciences and Medical Education, University of Southampton, Southampton SO17 1BJ, UK; Nick.Francis@soton.ac.uk

**Keywords:** peer education, qualitative, biology, students, antibiotics, antibiotic resistance, health education, AMR

## Abstract

Peer education (PE) has been used successfully to improve young peoples’ health-related behaviour. This paper describes a qualitative evaluation of the feasibility of university healthcare students delivering PE, covering self-care and antibiotic use for infections, to biology students in three UK schools (16–18 years), who then educated their peers. Twenty peer educators (PEds) participated in focus groups and two teachers took part in interviews to discuss PE feasibility. Data were analysed inductively. All participants reported that teaching students about antibiotic resistance was important. PE was used by PEds to gain communication skills and experience for their CV. PEds confidence increased with practice and group delivery. Interactive activities and real-life illness scenarios facilitated enjoyment. Barriers to PE were competing school priorities, no antibiotic content in the non-biology curriculum, controlling disruptive behaviour, and evaluation consent and questionnaire completion. Participation increased PEds’ awareness of appropriate antibiotic use. This qualitative study supports the feasibility of delivering PE in schools. Maximising interactive and illness scenario content, greater training and support for PEds, and inclusion of infection self-care and antibiotics in the national curriculum for all 16–18-year olds could help facilitate greater antibiotic education in schools. Simplifying consent and data collection procedures would facilitate future evaluations.

## 1. Introduction

Antibiotic resistance is related to antibiotic consumption at a national, local and individual level [1]. Many countries in the EU and more widely have a national action plan to reduce antibiotic use to help control this resistance [2]. There is a wide variation in antibiotic consumption between countries [3] and between communities in a country. This variation, such as between rural and city regions in Spain [4], suggests that differences in antibiotic use may in part relate to cultural differences [5]. A systematic review concluded that variability in antibiotic consumption may reflect inappropriate antibiotic use, even across similar settings [6]. The differences in antibiotic use between country and cultural settings, suggests that education at an early age may influence these culturally influenced behaviours, leading to a future generation with greater intentions to use antibiotics appropriately.

e-Bug is a global educational programme for young people aged 4 to 18 years that aims to educate and encourage appropriate health and hygiene behaviours to help prevent infections, and increase knowledge of antibiotics and vaccinations [7]. By improving knowledge and intentions of young people around hygiene, and antibiotic and vaccine use, the programme aims to help control antibiotic resistance through greater self-care of infections and appropriate use of antibiotics in the community [8]. The e-Bug programme includes lesson plans and activities for classroom use and student resources including games, quizzes, and revision guides [9]. The materials include a peer education (PE) resource, in which young people are encouraged to educate their younger or similar aged peers about infection prevention and antibiotics [10]. Peer-based programmes are effective because (i) they use role models to influence the majority through modelling behaviour or education, (ii) young people are most influenced by their peer group and people they find credible or with whom they identify with (peers in their own school), and (iii) social networks exist organically within peer groups, thus providing a natural communication channel for influencing change [11]. The e-Bug PE resources have been used to teach young people aged 9 to 14 years in schools or science shows [12]. The PE resources have also been used by young people to educate their younger peers, which has resulted in significant increases in student knowledge and confidence in both groups [10]. The e-Bug PE materials have not been evaluated in 16–18 year olds who actually may benefit the most, as they are beginning to make decisions about their own infections [13]. 

The aim of this study was to explore the views of student peer educators (PEds) and teachers about the feasibility of, and barriers and facilitators to university healthcare students delivering PE on antibiotic use, antibiotic resistance, and self-care for infections, to 16–18-year-old biology students, who then educate their school-year peers. This qualitative work was part of a larger feasibility study assessing recruitment and evaluation tools [14]. 

## 2. Results

### 2.1. Participants 

All the university PEds (six medical students in Cardiff; four pharmacy students in Manchester), and 10 of 11 secondary school PEds in the two participating Cardiff intervention schools participated in four focus groups. Two teachers, one from each intervention school in Cardiff, participated in telephone interviews. 

The overarching themes that were raised were views about antibiotic PE, facilitators of PE, barriers to PE, behavioural change and impact, and improvements to PE; representative quotes are given in Box 1, Box 2, Box 3 and Box 4 and Table 1.

### 2.2. Views about Peer Education 

School and university PEds, and teachers reported that antibiotic resistance is an extremely important health topic to teach to young people. University PEds also reported that being able to influence behaviours around antibiotic use in the local community was important. University and school student PEds reported that the lesson plan and materials were comprehensive and easy to use. Reported views about the benefits of PE for university PEds were mostly associated with promoting the university, professional development, confidence building, and skills gained. School student PEds reported that they could use their PE experience for university applications, personal statements and CVs, and teachers felt that this was a key incentive for students to volunteer. Teachers reported that school students with career interests that aligned with antimicrobial resistance (AMR), such as biology, science and health, were more motivated to participate in these extra-curricular activities. School PEds also reported gaining communication, teaching, time management, and planning and organisation skills. School teachers reported that the university students acted as good role models, and imparted useful information on their university courses. Teachers also reported that the teaching would enhance transferable skills, including communication skills, ability to work as part of a team, and focus independently on research. 

University student PEds from both universities expressed some anxiety about delivering PE to 16- to 18-year-old students. They had concerns about coming across as patronising to peers who were not much younger than them. Some students commented that they found giving the presentation was intimidating for them as educators as they were unsure whether the content that they delivered during PE was engaging enough or too basic for the biology learners. One student commented that it sometimes felt that they were more of a teacher than a peer, and that therefore there may be some resistance from students due to this. 

Box 1Views about antibiotic peer education.Importance of antimicrobial resistance Cardiff medical student (focus group four): “For me antibiotic resistance has been a big part of academic side of this year so being able to affect change in the local community has been really important.”Cardiff school biology student (focus group one): “I think it is a good thing because more awareness should be raised towards the misuse of antibiotics, so I think teaching young people this is really good.”Cardiff biology teacher (interviewee two): “… Antibiotic resistance is a massive topic, very important and very important to the NHS and trying to cut down on antibiotic use is something that that needs to be done and I think that they understand now, how this sort of model [PE] could help to do that.” Perceived benefits of the peer education resource Confidence: Cardiff medical student (focus group four): “And … that they [the university] can trust you to deliver it to a high enough standard and to solve, because you’re representing the university as well. …… They trust you to behave professionally, … so it is quite nice having that responsibility I think. I liked it.”Personal statement: Cardiff biology teacher (interviewee one): “I think that lots of the students thought it would be something that would look good on a personal statement…. Something that would help them if they went to an interview for a university or a job…where they’ve demonstrated skills.”Skills: Cardiff biology teacher (interviewee two): “They’re having to prepare lessons…. it is not just about learning knowledge they’re going to develop skills, aren’t they? They’re going to develop the ability of working as part of a team, where they have to work together to deliver something. They’re developing their communication skills, they’re developing their independent learning skills, they’re going to have to go away and do some further research, and there is a whole range of transferable skills that they were developing in this model.” Negative views about peer education Perceived to be a teacher rather than peer: Cardiff medical student (focus group four): “I suppose it felt being a bit more like a teacher at times than being a peer, which meant that there could be a bit of resistance from the students especially if they were giving up their lunch time or something.”Materials for biology students too basic: Cardiff medical student (focus group four): “I’m not sure that I was entirely convinced that they were the best people to target and I do not know if, at times I felt like I was being a little bit patronising because we’re so close in age to them. However, some of the teaching I thought was a little bit, it is not that long ago that I was an A level student, I think I would have felt a little bit patronised by some of the materials in it”.Manchester pharmacy student (focus group three): “However, when you do not know and you’re just constantly gauging, that is what I find a little bit intimidating. Because you look around and you’re trying to gauge from their expressions, are they engaging, are they switched off a bit, so that is the bit that I found intimidating rather than their age, if that makes sense?” Links to universities Cardiff biology teacher (interviewee one): “I think it is important to get the links between schools and the universities so that if the pupils are keen to go to university, that they get a taste of what is it is like in a sense and they get the opportunity to speak to … students at the university…”

### 2.3. Facilitators of Peer Education 

University and school PEds reported delivering the antibiotic lesson in groups of approximately three to five with different students delivering different components of the lesson. This peer support during preparation, practice and delivery of lessons was reported as the key facilitator for PE. School PEds reported that group delivery enabled them to reach a wide range of learners, allowed appropriate division of student tasks within the lesson, e.g., antibiotic presentation and a balloon-popping activity to demonstrate antibiotic resistance, and opportunity to receive feedback on lesson delivery from fellow PEds. School PEds reported that peer support came from friends participating in the lesson, who were able to encourage their peers to listen and behave. PEds delivering multiple lessons reported developing more confidence with each lesson, and felt better able to deliver lessons if they had prepared and practiced more beforehand. Teachers also described preparation and practice as being key to improving student confidence. Confidence was also related to students’ reported knowledge of the topic, with students who reported better knowledge about the topic expressing more confidence in the delivery. 

Interactive activities were reported as more engaging for school students by both PEds and teachers. All were very positive about the interactive activities on AMR involving balloon-popping, role-play and discussion. The inclusion of real-life and relevant examples and case studies was regarded as a way to increase engagement by all PEds. 

Box 2Facilitators of peer education.PE resource easy to use Cardiff medical student (focus group four): “As soon as we went to have our introductory meeting and given this pack and it had a lesson plan so it was easy to just go with it and you did not really have to think too much about what you were doing as such.” Peer support including peer education in small groups Cardiff medical student (focus group four): “However, also the students will respond better to different characters …. so, it is definitely good to have a range of people, I think, presenting it [PE].”Cardiff school biology student (focus group one): “I think as a group we try to help each other… I think that was really good because the person who you’re in a group with, they tell you what to say, what not to say, how to say it and then what to do as well…” Interactive teaching more engaging Cardiff medical student (focus group four): “Yeah everyone perks up when you pull these balloons out and they’re like, what are you going to do? And then when you pop them people are really excited which I get, I understand myself.”Cardiff school student (focus group one): “I enjoyed the lesson activities because it… got people interested in the talk and we kind of tested their knowledge and how, and if they actually understood what we were talking about.” Confidence and knowledge Manchester pharmacy student (focus group three): “Because this content [was what] we already covered in the previous years, it wasn’t [new] in terms of the knowledge, we were quite confident.”Cardiff biology student (focus group two): “Before I went I was actually very scared because I thought I was going to, it was going to be very bad, but then I found it much easier to teach too, because of that I think my confidence level was high and it helped me to do the other classes as well.”Cardiff biology teacher (interviewee two): “They were more confident in delivering the sessions the more that they had done, which was quite nice to see actually, they were not as scared as they were the first time that they did it.” Interactive and situational teaching Cardiff biology teacher (interviewee two): “I think that, the role play activities were quite good, because they get lots of misconceptions there. I think that [the role play] was something that not enough time was devoted to.”

### 2.4. Barriers to Peer Education 

A key barrier reported by school PEds was difficulties in controlling the behaviour of their peers during lessons. Some school PEds reported stress associated with teaching peers who were behaving badly and expressed that they were not respected as PEds. Some stated this was because they were a similar age, or marginally younger than PE participants, and therefore they found this was different from lessons their teacher might give, thus paying less attention. In one lesson observed by a researcher, there was no teacher present to supervise and PEds independently led the lesson, which may have contributed to some students misbehaving. School student PEds suggested that teachers should be present in the classroom to manage bad behaviour. 

In one Cardiff school, PEds reported that language barriers reduced the impact and effectiveness of the lesson, as English was not the first language for many students, reporting that learners had difficulty in understanding the lesson. School PEds reported adapting the lesson to other commonly spoken languages including Arabic. 

Other barriers reported by all PEds included the administrative elements of the consent process, as well as completion of knowledge questionnaires, which were necessary for quantitative data collection described elsewhere [14]. They reported that completing forms and questionnaires was boring and time-consuming, leaving less time for the activities. However, some PEds were understanding of these requirements, suggesting that they were a necessary component for research and evaluation. 

Barriers reported by the teachers were associated with their school demands and pressures. One teacher reported that parents and governors would not accept activities that were not based on the school curriculum during lesson time, therefore this school opted to run the PE lesson during lunchtime. A second teacher expressed concerns over finding time to observe PE lessons, as funds for a cover teacher were not available. One teacher reported that school student PEds lacked the ability to prepare and deliver the PE lessons independently. Students mostly replicated the lesson delivered by university students and did not suggest improving or changing the method, which may not have been as effective for them, as opposed to the university-student led lessons. 

Box 3Barriers to peer education.Behaviour of peers Cardiff school student (focus group one): “And they were like, why is she teaching us, she’s so much younger than us? And they did not listen and they started making fun [of me]. In addition, that made me feel really uncomfortable because I did not want to make them feel like [that]. I want to be a teacher. I’m trying to do something but because I was younger [by a few months] it made me feel uncomfortable.” Language barriers Cardiff school student (focus group one): “It was just because the last group that we did, they had a lot of language barriers, and most of them did not speak English very well. Although some of them speak my first language and then some of them speak like [names students] first language, which is Arabic.” Time and administration Cardiff medical student (focus group four): “It was a bit stressful at times because we’re trying to, especially with the consenting taking up so much time we were trying to fit a one hour lesson into 40 minutes or so”.Cardiff biology teacher, (interviewee one): “It costs for us to cover classes so then, I wouldn’t have been able to go and watch the lesson unless we had some kind of, some way of covering my class unfortunately.” Teacher pressures Cardiff biology teacher, (interviewee two): “Obviously the teachers are under pressure to complete the syllabus…. otherwise they would be receiving complaints that they’re not doing their job properly and our parents would write to governors.”

### 2.5. Behavioural Change and Impact

Some school and university PEds reported behaviour change around their own antibiotic use. Pharmacy students reported being more aware of antibiotic resistance campaigns and posters in public settings and having more conversations with friends and family about AMR. School PEds reported talking to family members about prudent antibiotic use and reported that participation in the study had influenced their views on antibiotics and they would now only take them if appropriate and necessary, such as for serious bacterial infections. 

Box 4Behavioural change and impact.Less likely to take antibiotics Cardiff school student (focus group one): “It’s also, doing this presentation has made me wary of taking any [antibiotics], so I do not think I’d ever take any unless it was a life and death situation.”  Awareness of campaigns Manchester pharmacy student (focus group three): “It’s actually made me notice a lot more of campaigning for antibiotic awareness as well. Once we did that PowerPoint presentation, afterwards I started to see signposts everywhere, adverts on the TV and you’re like, wow.” Motivation to discuss with family members Cardiff school student (focus group two): “I do not know, I’ve now had quite a few conversations and so on with family members about it.”Manchester pharmacy student (focus group three): “Even some of my family members are medical or veterinary practitioners and, some of the things they’re telling me about in their practice and what they do, I sort of question it… Therefore, doing this has made me a bit more aware of ways I can talk to other people when certain situations come up and say look, why are you doing that, actually question things as well.”

### 2.6. Improvements to PE 

There were several suggestions made by PEds and teachers to improve the PE lesson. Both PEds and teachers suggested that facilitating more group discussions within the PE lesson instead of using the formal PowerPoint provided could increase student engagement, discussion and learning. University PEds felt that the PowerPoint could have been improved by adding more real-life examples of AMR, and school PEds felt that the language used could have been made simpler for teaching peers, but that it gave a good overview of what they needed to cover. Both sets of PEds felt that the PowerPoint could have been improved if they were able to modify it to suit the group and their own individual teaching style to encourage discussion on important health concepts. Teachers recognised the PowerPoint as being very useful for teaching when they may not be very confident in delivering whole lessons but agreed that rather than replicating the lesson students should have taken the learning points and improved it. 

Biology PEds reported that they were given exactly the same lesson that they delivered to their peers, and this was rather basic for them; they would prefer more detail on antibiotics and resistance. One student mentioned: “*I think I would have preferred it if they used real case scenarios and explain more about how antibiotic resistance affects normal people in the real world, like how it can affect the ageing population*”, suggesting that the presentation was basic, and that more detail that is relatable would have been engaging. Other suggested improvements are in Table 1.

## 3. Discussion

This study identified that PE between university students and 16–18 year olds and from 16–18 year olds to similar aged peers is feasible. PE can be used to educate about AMR, as AMR is an important topic of interest. PE can help build student portfolios for university and work applications, and develop confidence and communication skills. All students favoured interactive activities, and were keen to learn more about the real-life impact of AMR using case studies. Practice and preparation for PEds was key to developing confidence for antibiotic PE lesson delivery. Support from schools and teachers is paramount to being able to carry out PE at a time that is convenient for the schools and students. 

Peer-based approaches to youth health promotion have been widely used, particularly for substance misuse prevention, sexual health and mental health education [15]. These areas of health promotion are pertinent because students may not wish to discuss drug, sexual and mental health issues with teachers. They have involved workshops with a similar interactive format to the antibiotic lessons given in this study or involve much more informal discussions with family and peers, that our PEds also reported. In one study, investigating peer-led teaching of sex education found that there was a progressive increase in knowledge about sexually transmitted infections and contraception, and behavioural changes such as decrease in risky sexual activity [16]. Other PE have shown similar benefits to our study with PEds gaining confidence. Students found preparing for and giving sexual health PE in groups was difficult if an individual’s opinions about sexual morality differed from another’s [15]. Differences in opinion about antibiotic use were not discussed as a problem by our PEds, but it would be worth exploring in future studies. Our quantitative findings suggest that this type of intervention can be useful to promote the prudent use of antibiotics, as seen in antibiotics knowledge gain [14]. 

In a previous study [17], students who were regarded as influential, good communicators, and good role models were selected to be PEds. In the present study, the PEds were university students taking extra modules, or school biology students. The university students were seen as good role models for the biology students, and the biology students in turn had good knowledge of antibiotics and infection prevention as tested within the questionnaire [14], which they reported facilitated their confidence and knowledge transfer to non-biology peers. Antibiotics PE was used in Moldova in which 12–13 year olds were trained to teach their peers and parents that antibiotics should not be used to treat colds and flu [18]. The Moldovian study included 34 PEds giving a series of lessons with over 1700 students without their teachers present; participants attended six PE sessions each and student misbehaviour was not reported as a problem. The importance of thorough preparation for the PE was raised in all the PE reports [15]. The Moldovian student PEds had two days of training and taught at least 10 sessions each; other non-antibiotic PE projects also included several sessions of training, and PEds usually gave several PE sessions [15]. Training and experience probably increases confidence to teach fellow students, reducing participant behaviour problems. In our study, PEds only had two to three hours preparation, which is much less than in previous projects. School (not university) PEds reported that student misbehaviour hampered the lesson, but this difficulty participating in PE decreased with school student practice. A recent study investigated peer-led team learning and one research question involved group dynamics and managing different student personalities [19]. Their findings included establishing ground rules, balancing personalities by reining in dominant students, encouraging younger students to speak up, making an active effort to ensure that everyone speaks at least once, and using turn-taking to answer questions. Incorporating methods such as this in future studies may help reduce students misbehaviour and help maintain healthy group dynamics [19].

Ideally students need greater training and PEds should have thorough lesson practice and preparation to maximise knowledge of the subject matter and improve lesson delivery. However, this thorough preparation may be very difficult in the busy school curriculum and without additional teacher support. University students did not report school student misbehaviour as a problem for them. This could have been because of greater maturity, confidence, and/or skills. Alternatively, the biology students may have respected and related to the university students, as they had future aspirations for careers in science, and as a result may have felt more engaged in the project. The ethos of PE is that students deliver PE unsupervised, but this may be necessary for antibiotic health education, and teacher supervision may decrease student misbehaviour. Teacher supervision should be discussed in future projects if the training provision for school PEds remains at two to three hours. An alternative strategy in some schools would be to ask university PEds to educate both the biology and their non-science student peers; however, this would remove the school student PE aspect and its reported benefits. Another PE study confirmed that increase in confidence of PEds was an important outcome and benefit of PE, especially when students with varying levels of ability were chosen to be PEds [10].

Previous evaluations of educational resources around antibiotic use have also confirmed that participants prefer to discuss clinical scenarios [20], rather than didactic talks given through PowerPoints. In the Moldovian evaluation, the intervention included professionally filmed videos of students acting out different illness scenarios, and each of these were then discussed by the group [18]. Tailoring the materials to different groups of abilities is also important to ensure that students are engaged with the content and it is age-appropriate. Learners do better if the lesson and activity is matched to their ability, and our biology student PEds commented that the content was quite basic for them. Factors including age, ability, and self-efficacy impact adult education participation, and as a result, intentions to learn, and actual learning [21]. This suggests that ability needs to match the content of what is being taught to encourage learning to take place, and improve self-efficacy—to feel capable of learning, and motivated to learn. 

### Strengths and Limitations

This is the first study that the researchers are aware of exploring young peoples’ opinions of delivering PE around AMR and antibiotics. Although this was a feasibility study, we interviewed 20 PEds and achieved data saturation for the university and school PEds. Due to lack of time available in schools, the study only involved two teachers who gave consistent data, and none of the school students taught by their peers; it would be useful to obtain views from all groups in future PE evaluations. The Manchester school students did not deliver PE, which limits the number of school settings from which views were collected. However, this limitation is an important part of any feasibility study and indicates that forward planning is needed for schools to participate fully, and stresses the need to avoid busier times in their school calendar such as exam periods. 

## 4. Materials and Methods 

### 4.1. Study Design

A qualitative study investigating the feasibility of a peer-education evaluation using focus groups and interviews with teachers and students.

### 4.2. Setting

The project was undertaken in a convenience sample of secondary schools close to the two participating universities (Cardiff University and University of Manchester). 

### 4.3. University Student, School Student and Teacher Recruitment

Using convenience sampling, university pharmacy (Manchester) and medical (Cardiff) students, were invited to be trained as PEds around AMR. Five of a convenience sample of 17 schools approached in Cardiff and Manchester agreed to participate, and were randomly assigned to either the PE intervention (three schools) or usual lessons (two controls). The full methods describing the setting and school recruitment for the feasibility study are discussed elsewhere [14]. Six to twelve weeks after the PE lessons, PEds were invited by email (university PEds directly, and school PEds through their teacher), from the local researchers to participate in student focus groups, held either at the participating universities, or the participating schools. School teachers were emailed and invited to participate in telephone interviews to discuss the project and PE process.

### 4.4. Peer Education Intervention

School students in Cardiff gave informed consent at the beginning of the first lesson to participate in the study including receiving one text per month over three consecutive months asking about their antibiotic use. Parents of students in both Cardiff and Manchester schools (where texts were not used) could opt for their child to not participate. In each school, biology students were taught an e-Bug antibiotics lesson by university student PEds. These biology school students were then invited to volunteer to be PEds and deliver the lesson in groups of three to five to their peers. University student PEds were trained by a member of the research team and biology students were trained by either one of the researchers, or university students. PEds were encouraged to practice lesson delivery and content. PE lessons in intervention schools were observed by a researcher. Due to the timing of the study being close to exams, school student PE was only carried out in Cardiff schools, while Manchester school students only received PE from university students. 

### 4.5. Materials

A common e-Bug antibiotics PE lesson plan was developed for use in all university-led and 16–18-year-old led antibiotics PE lessons, to ensure consistent messages and learning (Appendix A). This included background information for the educator, a checklist of key words and learning outcomes, a PowerPoint presentation of antibiotic resistance, and an interactive demonstration of an antibiotic-resistance activity using balloon popping, scenarios about young people with common infections, and right or wrong statements about antibiotic use in young people, for discussion. PEds were also provided with answer sheets for scenarios and right or wrong statements to facilitate discussion during the lessons. A questionnaire was also developed to assess antibiotics knowledge before, immediately after the lesson, and three month later. (Appendix A). 

### 4.6. Ethical Considerations

Teachers and school PEds provided consent to participate and for data to be transcribed verbatim and anonymised quotes to be used in publications. The study gained local ethical approval from PHE Research Ethics Governance Group. (Ethical code: R&D 340).

Focus groups and interviews were facilitated by trained qualitative interviewers (CB and RS) and lasted 27 to 53 min. Focus groups and interviews were audio-recorded, and researchers took relevant notes during data collection. Immediately after the interviews and focus groups, the researchers made notes of the major themes arising. Audio recordings were transcribed verbatim and transcripts were checked for accuracy against recordings by a researcher. Data were analysed using NVivo Pro 11 qualitative analysis software (QSR International). An inductive analysis approach was used, in which all transcripts were examined for themes by one researcher (CB) and 25% were independently coded in parallel by a second researcher (VB). Categories and themes were then discussed, combined and refined between researchers until agreement was reached. The transcripts were then further analysed by the primary researcher. Themes were discussed by the whole research group and representative quotes were agreed upon to demonstrate each theme.

## 5. Conclusions

This study found general support for PE around antibiotics, and evidence that it is likely to be feasible, as well as data on barriers and facilitators that will help in the development of a larger study to assess the effects of such an approach. Future PE and evaluations should provide PEds with substantial (probably half a day or more) training and practice, and peer and teacher support. PE needs to be adapted for each school’s needs and support including single tier PE (just involving university students) or two-tier (university to school then peer to peer within schools) if practical and feasible within the school lesson timetable. The lesson plans should continue with the interactive activities, relevant student illness scenarios with a PowerPoint and activities that can be adapted for student ability. In future studies, the questionnaire and consent process should be completed at a different time to the intervention lesson. As teachers reported that schools would be more motivated to promote antibiotics PE if the topic was part of the national curriculum, the Department of Education should be lobbied to make prevention and self-care of infections and antibiotics an obligatory part of the national curriculum for all 16–18 year olds in education, not just those studying biology/science subjects.

## Figures and Tables

**Table 1 antibiotics-09-00194-t001:** Improvements suggested by students and teachers for the peer education programme.

Lesson Component/Barrier	Suggested Improvements from Students and Teachers	Implications/Rationale
Difficulty of lesson too basic for some biology students	The lesson activities for biology students should go into greater depth to match their abilities.	Biology curricula for A-Level students includes more in-depth information, therefore this will align with the curriculum and add to students’ knowledge.
PowerPoint activity perceived to be boring	Replace PowerPoint with the e-Bug YouTube animations showing bacteria developing and spreading antibiotic resistance.	More interactive visual demonstrations will be engaging for students and may help with knowledge gain and understanding.
Antibiotic resistance PowerPoint presentation not relevant to students	Presentation could include information on real-life case studies and impact of AMR to increase motivation to self-care and reduce antibiotic use.	Students will be better able to understand the impact of AMR in the real world and will put this into context that AMR is an issue that affects us, but there are ways that we can self-care to help reduce the spread of antibiotic resistance.
Activity 1. Although fun, demonstration of AMR using balloons too simple for some	For biology students, include greater detail on how antibiotic resistance can be passed on to different bacteria, and how resistant bacteria can replicate.	To match A-Level curriculum, and enhance learning by university PEds.
Activity 2a. Scenarios involving young people, self-care and antibiotics could be improved	Encourage role-play during scenarios so that students can actively get involved.	Role-play is interactive and encourages students attending the lesson to actively play a part, therefore may help in knowledge acquisition and retention, both in participants and PEds.
Activity 2b. Antibiotics Right or Wrong	Use e-Bug debate kit scenarios to discuss antibiotics and their use, would be a useful alternative activity.	Debate kit activities may be suitable as it encourages discussion between peers, and allows peers to argue different standpoints regarding AMR.
Self-Care TARGET Respiratory Tract Infection antibiotic leaflet not covered sufficiently	Increase time during the lesson to discuss the leaflet.	If the leaflet is discussed during lesson time students will be able to give their feedback on it, suggest improvements and discuss how they plan to introduce it to their families.
Participants misbehaviour	Greater training of PEds or supervision of lessons by teachers, but lack of teacher time.	Greater training in or out of school hours, by university PEds. Teacher supervision of PE until PEds are very confident. Single tier PE by university students to all 16–18 year olds.
Administration for evaluation reduced the available time for the lesson activities	Undertake outside the actual PE lesson.	Gain student consent for any formal evaluation outside the lesson. Deliver questionnaires at a separate lesson or electronically.
School governors and therefore teachers not motivated to include PE on antibiotics	Include in National Curriculum.	Approach school via school governors, until included in National Curriculum.

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
