# Peer review of "Teacher and Student Views on the Feasibility of Peer to Peer Education as a Model to Educate 16–18 Year Olds on Prudent Antibiotic Use—A Qualitative Study"

_antibiotics, 2020, doi:10.3390/antibiotics9040194_

Round 1
Reviewer 1 Report
The authors describe a qualitative evaluation of the feasibility of university healthcare students delivering PE, covering self-care and antibiotic use for infections, to biology students in three UK schools (16-18 years).
In my opinion this paper have not a scientific sound. The methods used did not respect any basic principles of th educational simulation. I don't think that this report can be add some relevant scientific information to the actual knowledge. The methods of interaction are flaw
The are not clear the results and the discussion is too speculative.
Reviewer 2 Report
As you mentioned since the beginning, the study is qualitative. Anyhow, my question (concern) is: did you do some quantitative investigation in order to support your conclusions? Are some data analysed in this respect?
Is it possible to put as ‚supplementary materials’ examples of the ppt. presentation / presentations in order to help us to clarify the matter?
Reviewer 3 Report
McNulty et al. conducts a study for effectiveness of the peer education on antibiotic resistance among teenagers in schools and universities. Authors provided the content of the course and reactions of the students. Several aspects while conducting and concluding the study has been summarized or presented as quoted text which are mostly responses of individuals. Authors conclude the some of the limitations and assert that the antibiotic resistance should be a part of curriculum.
Overall, I found the content of the course informative simple and can be effective. Study is Although I have following concern that should be addressed:
- Too much quoted text in boxes, which is defeat the purpose of study and it is better if there are bullet points concluded by the author that describe the responses in general.
- Author conclude that the antibiotic resistance and awareness would be a part of the curriculum, but how and at what level? Should it be limited to biology courses or can be included in other curriculum that reaches wider audience outside the students studying biology.
Reviewer 4 Report
In this article, a qualitative study is carried out on the feasibility of a peer to peer education model on prudent antibiotic use in students between 16-18 years of age.
The introduction is well-founded and complete.
The results are fully stated and I think it is adequate.
The discussion is correct. The authors comment that peer-based approaches to health promotion have been widely used for substance misuse prevention, sexual health, and mental health education. Authors say that these areas are pertinent because students may not wish to discuss these issues with teachers. But authors do not mention the effectiveness of these interventions. Perhaps it would be right that the authors discuss whether this type of intervention can be useful in promoting the prudent use of antibiotics. Although the efficacy of the intervention is not the main objective of the study, reference could be made to this subject.
I think it would be good that the authors detail the methods used a little more since it refers to an unpublished study yet, and it is difficult to evaluate the methods used.
